# Applying qualitative methods to experimental designs: A tutorial for the behavioral sciences

**Hidde Jelmer Leplaa** [ID][¤]*, **Jari A. Tönjes, Mariska Bouterse, Karlijn F. B. Soppe, Irene Klugkist**

Department of Methodology and Statistics, Utrecht University, Utrecht, The Netherlands

¤ Current address: Faculty of Social and Behavioral Sciences, Utrecht University, Utrecht, The Netherlands
* H.J.Leplaa@uu.nl

**Data availability statement:** The authors of the qualitative study, the worked example in our

## Abstract

Studies with experimental designs are almost invariably evaluated with quantitative outcomes and methods, both in behavioral sciences and other disciplines. We argue that there can be added value of using qualitative methods for the evaluation of (behavioral) experiments. Incorporating qualitative data can enhance the ecological validity of a study, by acquiring a more holistic understanding of the phenomenon of interest. There is, however, little methodological guidance on how to implement such an approach.

In this paper we present the different steps and considerations for a qualitative evaluation of results in experimental designs. Methodological guidelines are offered for each stage of a study, from formulation of the research goals, through data collection and data analysis, to the interpretation of a potential effect of the intervention. In addition, there is ample attention for ensuring the rigor of the research. The presented guidelines are developed and illustrated using an empirical example, in which a constructivist grounded theory approach was applied to evaluate the effect of empathy prompts on the motivation to adhere to COVID-19 regulations.

## Introduction

Experimental designs are common research designs in the social and behavioral sciences [1,2]. Although there are different types of experimental designs, shortly outlined hereafter, they almost invariably have one thing in common: quantitative data are used to evaluate the effects. In this paper, we demonstrate and discuss the use of qualitative data to evaluate experimental designs, resulting in a methodological tutorial which may be especially beneficial for studies aiming to explain human behavior.

Before elaborating further on the different, qualitative and quantitative, approaches to measure the potential effect of an experimental manipulation, we shortly recap the different experimental designs, starting with the distinction between true experiments and quasi-experiments [3]. In both true and quasi-experiments participants are subjected to some sort

paper, have decided to make the data openly available. We have added the relevant information in the submission, with a link to the repository (https://osf.io/e4zfw/).

**Funding:** The author(s) received no specific funding for this work.

**Competing interests:** The authors have declared that no competing interests exist.

of treatment or manipulation, and the effect of this manipulation is measured on the outcome(s) of interest. In a quasi-experiment participants are naturally members of one of the experimental conditions, without the possibility for the researchers to control the allocation of participants. In true experiments there is full randomization, that is, the participants are randomly assigned to one of the conditions of the experiment. Random assignment enhances comparability between groups as it diminishes the effect of confounders and, therefore, positively affects the internal validity. Although quasi-experiments can lead to undesired differences between conditions, in practice full randomization is not always possible. Throughout the paper, when we use the term 'experiment' or 'experimental design' this includes both true and quasi- experiments.

A different typology of experiments is the distinction between lab, field, and natural experiments [4]. Laboratory experiments are conducted in highly controlled environments, where researchers manipulate one or more independent variables to create the conditions and subsequently measure the effect of the conditions on the outcome of interest. While laboratory experiments have high internal validity due to the high level of control, they typically have low ecological validity, because the lab situation may be different from daily life [5]. Alternatively, field experiments are conducted in the everyday environment while still allowing the researcher to manipulate the experimental conditions. Therefore, in field experiments there is less control over extraneous (confounding) variables, potentially lowering internal validity, but have higher ecological validity [6]. Finally, in natural experiments there is no manipulation controlled by the researcher at all, instead naturally existing groups, that is, conditions, are compared on the outcome of interest in everyday life. Although this can provide valuable, practical information with high ecological validity, there is a higher risk of confounding variables.

Irrespective of the type of experiment the researcher must define the intended outcome, that is, the change as is defined or measured within the study that is assumed to be the consequence of being exposed to an intervention or manipulation. We will refer to the intended outcome throughout this paper as the "*intervention effect*", a term that can refer to quantitative or qualitative outcomes.

In quantitative research the outcome is a measure that reflects the construct intended to be influenced by the intervention. This is generally done by providing operationalizations of the key constructs of interest as quantitatively measurable variables, that are subsequently analyzed by quantitative methods. Using the empirical-analytical paradigm, common in behavioral sciences, one assumes that the universe can be explained through researching empirical data, where a condition for the research is that it is reproducible, replicable, systematic, and transparent [7]. To achieve these goals a form of reductionism is implied.

Reductionism in research is the process of simplifying a situation to scores on a small number of variables [8]. Since human behavior is intricate, researchers can engage in using reductionism by bringing complex human beings down to scores on variables. Through reductionism it is feasible to understand if some variables are related, however, quantitative research seldom explains the exact process under investigation. To fully understand human behavior one may need to study the larger context, that is, the complex human being as a whole. Research that focuses on small segments of reality, as is done with quantitative analyses and reductionism, may lack ecological validity by not considering all aspects of the real-life situation and the complex human being of interest.

We therefore argue that it can be beneficial to utilize a more holistic approach for the evaluation of results in experimental designs, especially in the context of behavioral sciences. Holistic approaches are typically associated with an interpretist point of view, and regularly apply qualitative methods [9]. It builds on the idea that human beings and their behavior and

motivations are complex and thus need to be studied without reducing them to some numerical variables. Adopting a more holistic view enables researchers to more fully understand the phenomenon of interest and as such can positively influence the ecological validity of a study [10]. Next to the holistic approach, qualitative research is often associated with an inductive goal, i.e. theory forming, which can be achieved through an abductive approach [11]. Abductive processes are by nature iterative, as they follow sequences of formulating preliminary explanations based on data which will be tested within the same study with new data. The use of qualitative methods is rather common in several fields of research, and there is much literature available discussing the ways to conduct qualitative research e.g. [12–15], with a clear focus on idiographic knowledge: thoroughly understanding the individual. Since understanding the individual is the focus of much qualitative research, there is little literature on using a qualitative approach specifically for the evaluation of experimental designs in behavioral science.

Some papers offering guidelines for qualitative evaluations of experiments are found in humanities [16], management [17], and political science [18]. To the best of our knowledge, there are no methodological papers in the field of behavioral sciences, but there are a few examples of studies that apply qualitative methods for the evaluation of experiments [19–21], and often they quantify the qualitative results [22,23]. These studies offer little to no methodological guidance for other behavioral researchers to apply qualitative methods in their experimental studies.

With this methodological tutorial we aim to provide such guidelines. We provide a step-by-step approach for future researchers aiming for a qualitative evaluation of an experimental design, supported with an example. This worked example describes a field study that investigates the effect of empathy-inducing interventions on the willingness to adhere to social distancing rules during the COVID-19 pandemic. De Ridder et al. [24] set-up an experimental (quantitative) study for this goal, that provided the framework for another study executed independently from de Ridder (Glebbeek et al. [25]; from here on referred to as GLS). GLS used qualitative methods to evaluate the effect of the intervention following a constructivist grounded theory approach and this study is used as a worked example to develop and illustrate the general methodological guidelines presented in this paper.

The rest of the paper is organized as follows. First, we shortly describe those aspects of the study of De Ridder et al. [24] that also apply to the qualitative study by GLS, that is the design and intervention, followed by the motivation for the GLS study. In the next section, we present the methodological guidelines for qualitative evaluation of an experiment, structured in four stages of the research process: defining the research goals, data collection, data analysis, and determining the intervention effect. A separate subsection describes an overarching fifth stage, which partains the process of reflecting and adjusting during the other stages. The paper concludes with a discussion of the merits of (additional) qualitative evaluation of experimental effects compared to a purely quantitative analysis. Finally, we provide recommendations for researchers also considering a qualitative evaluation of an experiment.

## 1 Context of the worked example

In 2020, in the Netherlands, De Ridder and colleagues investigated the effect of an empathy-inducing intervention on the willingness to adhere to the social distancing rules during the COVID-19 pandemic, using a quasi-experiment. One of the measures taken by the Dutch government during the pandemic was social distancing, which included keeping 1.5 meters distance between people.

The field study employed an A-B design [26] lasting for 6 weeks, with three sequences of a control week (A) and an experimental week (B) at three campus locations (square outside college hall, main entrance lecture hall, and entrance lecture rooms). The same approach, but with four instead of three sequences of A-B weeks, thus 8 instead of 6 weeks long, was executed a second time four months after the initial wave. In the data-analysis the researchers assumed this to be a between-subjects design, though realistically participants could have been present at multiple occasions throughout the weeks, but without personal identification this could not be detected.

The intervention consisted of prompts to induce empathy and contained three elements: (1) a social robot encouraging people to keep distance; (2) posters of student and staff models with a text expressing a prompt for empathy-based distancing (e.g., 'I have asthma. Keep your distance for me'); and (3) a rail of movie clips of the same models used in (2) with the same texts shown on screens. To determine the effects of the empathy prompts for promoting distancing, De Ridder et al. [24] used the actual distance between people which was measured using camera recordings. The results of the quantitative study can be found in the paper by De Ridder et al. [24].

Whereas the camera recordings provided quantitative data on the actual behavior, it does not provide insight in reasons or motivations to (not) adhere to the social distancing rules. To further investigate if and how the intervention had an effect on those aspects, GLS added a qualitative inquiry using the context of the main study. Certain aspects of the qualitative study were thus set in stone, that is, the intervention and the data collection in the two waves of A-B sequences were predetermined. All other choices were open, providing a good playing field for our study, that is, investigating how to perform a qualitative evaluation of results from experimental designs in the context of behavioral research. While the GLS study is an example of a quasi-experiment, the lessons and guidelines can be applied directly to true experiments as well. Note that the details and empirical results are reported in Glebbeek, Leplaa, and Soppe [25], while we use the qualitative study as a worked example to derive and illustrate the methodological guidelines presented in this paper.

The studies of De Ridder et al., and GLS as an addendum, were approved by the Ethics Committee of the Faculty of Social and Behavioral Sciences, Utrecht University (file number 20-479). Our methodological tutorial was exempt on the basis of not collecting nor handling data ourselves. We did however have access to the data for research purposes from 2019 through 2023, while the data were anonymized so it could not be brought back to individual participants.

## 2 Methodological approach

In this section, we provide a step by step approach developed and illustrated by the GLS study. We explain the methodology followed in GLS, the rationale behind it, and the lessons learned throughout the study.

To ensure transparency and rigor in research, it is common practice to preregister studies and make data open. Also, for the context of qualitative evaluation of experiments these are important considerations before executing the study.

In the context of qualitative studies, preregistration is a debated issue with proponents (e.g. [27]) and opponents (e.g. [28]). Although the discussion of compatibility of qualitative research and preregistration is beyond the scope of our study, GLS decided to preregister their research to formally document their intended approach before starting the study. The preregistration document can be accessed on the Open Science Framework: https://archive.org/details/osf-registrations-knb2d-v1.

Open data is also an essential aspect of scientific research as it enhances transparency, accountability, and reproducibility. However, the application of this practice in qualitative research is also debated [29,30]. It is argued that openly sharing qualitative data may result in a loss of context and can compromise confidentiality [29]. Despite the impossibility to perfectly interpret the data due to the loss of context, GLS decided to make the data publicly available through https://osf.io/e4zfw/ for the purpose of transparency.

After addressing the considerations of preregistration and open data, the actual research can begin. In Fig 1, we outline the stages involved in executing a qualitative evaluation of results in experimental designs. While the first stages of defining the research goal, collecting data, and data analysis are common to all empirical studies, the vertical stage labeled 'reflecting and adjusting' is more characteristic of qualitative research and is essential to its success. The arrows pointing upwards on the right-hand side of the figure highlight an essential aspect of qualitative research - the need for reflection and adjustment throughout the research process. It is common in qualitative research to revisit previous stages. This not only improves the quality of the data but also enhances the final analysis and conclusions drawn from the study, including establishing the intervention effect, as shown in the final stage at the bottom of Fig 1.

In the following sections, we will provide a detailed description of each stage shown in the figure. We will highlight general considerations that are important at each stage, as well as the specific choices made in GLS.

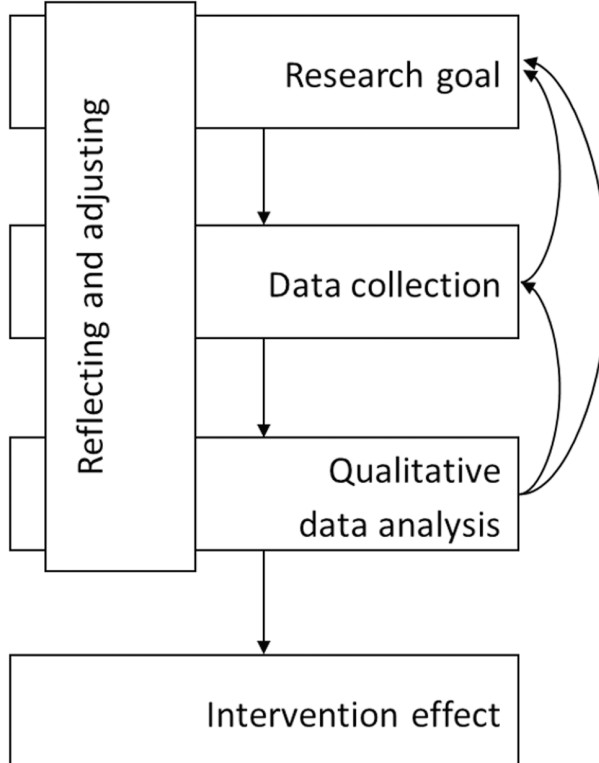

**Fig 1. Visualization of the methodological approach of a qualitative experimental study.**

## 2.1 Defining the research goals

In order to conduct a successful qualitative evaluation of an experiment, it is crucial to define its goals explicitly. The open and iterative nature of qualitative research makes clear goal-setting even more critical, as researchers often move back and forth between various stages, which can lead to *scope creep*, that is, a gradual shift away from the original research goals [31]. Therefore, formulating unambiguous goals and regularly reviewing them can help maintain the intended focus, or shift the focus by updating the goals in the desired direction.

Defining clear research goals is particularly important in studies of intervention effects and requires a clear description of the intervention under investigation and the outcome of interest, including how, when, and where the effects are being examined. In some studies intervention effects could be evaluated exclusively through qualitative data: researchers could decide that the qualitative methods provide all the required information to answer the research question. In other studies, intervention effects could also be studied with the qualitative data as an addition to quantitative data within the same study. The latter situation requires that the qualitative component adds value to the quantitative study, for instance, through triangulation or complementing [32].

In both scenarios, the qualitative investigation of potential intervention effects allows for a more comprehensive understanding of behavior and motivation compared to quantitative research [8,10]. This benefit can however only be achieved with in-depth knowledge of the field.

Therefore, a thorough review of existing literature is required to frame the study and ensure its relevance [33]. Familiarity with leading theories and concepts in the relevant field can lead to the identification of *sensitizing concepts*, which are concepts found in the literature that could be elements of the answer to the research question [13]. Sensitizing concepts can be used to design the study, including data collection and analysis, and can be valuable in defining the research goal. While these concepts are not final, as the definitive concepts and theoretical model will be drawn from the data, sensitizing concepts can guide the study towards achieving its goals.

GLS provides an example of adding a qualitative element to a quantitative study. While the quantitative study focuses on the actual distance between people, which is just one aspect of behavior, the qualitative study aims to provide an understanding of more aspects of behavior and uncover the reasoning behind it. *Empathy* was identified as sensitizing concept and the literature search showed that *empathy* comprises multiple aspects, these are emotional simulation, perspective-taking, and emotion-regulation [34]. Understanding these different facets is crucial for being sensitive to the various ways in which *empathy* can manifest in the study.

The research goal GLS defined for the qualitative study was to achieve a holistic understanding of both the behavior and the motivations for this behavior of participants in the context of COVID-19-related measures. Specifically, they aimed to identify the strategies that participants employed to adhere to the COVID-19-related measures and the motivations behind their choices. This allowed them to test the underlying theory on the effectiveness of prompts, with a predominantly inductive approach instead of a more narrow assessment of the behavioral outcomes.

## 2.2 Data collection

In qualitative research, just as in other studies, data collection needs to be carefully planned to ensure that the chosen methods generate data that can effectively address the research question. It is both advisable and considered a standard practice in qualitative research to initiate data analysis as soon as part of the data becomes accessible. This proactive approach not only

encourages critical reflection on the research process but also facilitates timely adjustments to data collection methods when needed. Typical data collection methods in qualitative research are observations, interviews, focus groups, and archival data [35]. It is important to note here that in the context of an experiment, the content of the intervention might limit the options for plausible data collection methods. Perhaps the most common methods, and what GLS used in their study, are observations and interviews. Therefore, we will focus on discussing the strengths and limitations of these two methods.

Observations are particularly valuable as they enable the study of naturalistic behavior and covert field observations can achieve a high level of ecological validity (e.g., [36,37]). Two types of observational studies are distinguished: systematic and non-systematic. Systematic observation involves using a pre-defined checklist to quickly and accurately record behavior with high inter-rater reliability [38]. However, it may not offer the holistic understanding, since what is being observed is determined beforehand. In contrast, non-systematic observation allow researchers to capture all relevant aspects of the situation in their field notes, which provides a detailed description of the behavior and its contextual setting. This approach can result in a more thorough understanding of the behavior and the factors that contribute to it. However, non-systematic observations are more prone to observer effects, that is, lower inter-rater reliability [37].

Interviews are a valuable research tool for gathering detailed information on the experiences, rationales, or attitudes of individual people [39,40, e.g., ]. While interviews may not provide information on naturalistic behavior, they can offer insight into intentions and experiences. Two main types of interviews are distinguished: open and semi-structured. In open interviews only a starting topic is selected and the interviewer and interviewee discuss what seems relevant to them. Open interviews can provide a holistic understanding of the individual, allowing the interviewee to address topics the researchers may not have considered. However, the lack of structure in open interviews can lead to some topics being overlooked, creating differences between interviews, which is an example of an interviewer effect that may limit the quality of data obtained [40].

Semi-structured interviews on the other hand follow a more predictable pattern by predetermining to a greater extent the content of the interview, which enhances comparability across interviews but may limit exploration [40]. To structure such interviews and ensure comparability, it is recommended to use a topic list that includes the topics to be discussed, their order, and specific wording of questions [41]. Such a list can help focus on relevant information and reduce unwanted researcher bias [42]. It also enables researchers to ask relevant questions that are more likely to elicit valuable data on the relevant concepts.

In qualitative research, one common practice is to employ method triangulation, which involves using multiple types of data within a single study [43]. By using method triangulation, researchers can study the phenomenon of interest from multiple perspectives, which ultimately enhances their understanding of it. Method triangulation allows researchers to choose data collection methods that complement each other's strengths and weaknesses. For instance, while interviews may provide insight into intentions, observations may provide information on actual behavior.

In GLS, the researchers utilized both systematic and non-systematic observation methods to investigate whether individuals adjust their behavior in response to prompts and to identify circumstances in which individuals adhere to or violate COVID-19-related measures. For systematic observations, a simple checklist was provided to record whether individuals seemed to notice the prompts, kept their distance from others, and if they walked alone or in a group (recording group size). Non-systematic observations were included through field notes, that

provided a more detailed description of individual behavior, their relationship with potential companions, and changes in behavior related to adherence to COVID-19-related measures. Participants were informed about data collection through signs with explicit information about the research. In addition the project's purpose and procedures were thoroughly explained on the university website.

Six researchers conducted these observations over a seven-week period in the first wave of data collection. Assigning the same task to multiple researchers is known as researcher triangulation and will reduce the influence of individual preconceptions [44,45]. A pilot week was used to determine the observation schedule and test the checklist, followed by three A-B iterations. The A-weeks served as the control condition with no intervention, while the B-weeks were the experimental condition. Topic lists for both weeks are included in Appendix A and Appendix B. A total of 52 hours of observations were conducted for the control condition, and 70 hours for the experimental condition, observing a total of 1002 individuals. The sample consists predominantly of highly educated adolescences, since the study location is almost exclusively visited by students and employees of the university and the university of applied sciences.

In addition, GLS used method triangulation through short interviews that complemented the information from the observations. The same six researchers conducted 5-minute interviews while walking with the people they observed, in the last week of data collection of the first wave. The researchers used a topic list to guide the conversation aiming for information about the motivation for their behavior, added in Appendix C. A total of 42 participants were interviewed, equally split between the control and experimental group. The interviews were transcribed intelligent verbatim to ensure the preservation of meaning [46].

The second wave of data collection was announced at a later stage and took place half a year after the first wave. The duration of the second wave of the study was eight weeks in total and consisted of four sequences of a control and an intervention week. The intervention was the same as in the first wave and was also placed at the same locations. The second wave enabled GLS to collect the data needed to answer the research question and the design was informed by the results from the first wave since data analysis was already conducted for that part of the data. Both the short and the long interviews were preceded by asking respondents for consent to record the conversation and to use the collected data for scientific publication. Their consent was also audio-recorded.

The key objective for the qualitative data collection in this wave was to gain a better understanding of the motivations and strategies of participants. To obtain more in-depth data, the researchers conducted longer interviews (approximately 45 minutes) with a predefined topic list to enable a comprehensive understanding of the processes that potentially influenced the participants' behavior. The topic list was based on the data collection of the first wave and contained topics like *Social life*, *Home situation*, and *Study-related contact*. For each topic, a starting question was predefined (e.g., 'Can you describe your recent interactions with fellow students and lecturers?') as well as some follow-up questions (e.g., 'How does this differ from before corona?') and subtopics (e.g., *agreements*, *activities*, *group composition*). The topic list is added in Appendix D.

The interviews in the second wave of data collection were done by five researchers at one of three locations: a building near the interventions, another university building away from the interventions, and online. Of the 39 participants, six were interviewed in the intervention-weeks and near the intervention location and were therefore considered part of the experimental group. The control group consisted of 33 participants who were interviewed online (3), on a location away from the intervention (13), or near an intervention location but during

a control week (17). The participants of both types of interviews were all students at the university or the university of applied sciences. Besides three non-Dutch participants in the control group, all participants were Dutch natives. While participants could have been the subject in multiple observations, no participants were interviewed more than once. The interviews were transcribed intelligent verbatim in order to stay true to the words of the respondent.

By using non-systematic and systematic observations as well as short interviews and semistructured interviews, GLS aimed for a comprehensive overview of the potential effect of the intervention. By combining data from these different sources, they expected to be able to describe the relation between behavior and underlying motivations, attitudes and intentions with respect to COVID-19-related measures and compare results between groups with and without exposure to the intervention.

As stated earlier, it is common practice to start initial data analyses simultaneously with collecting data in qualitative research. This can offer valuable insights and enables researchers to improve their data collection methods or identify confounders not anticipated beforehand. Before we discuss the qualitative data analysis in the next section, we first elaborate on the process of reflecting and adjusting that is typical in qualitative research and is needed to ensure the quality of both the collection and the analysis of data.

## 2.3 Reflecting and adjusting

Maintaining rigor throughout the research process is critical to ensuring the overall quality of the research. Four critical aspects of rigor include dependability, confirmability, transferability, and credibility [43,47]. Dependability and confirmability emphasize the importance of minimizing the researcher's influence [48]. Rigorous studies should produce objective findings that are not influenced by the researcher's preconceptions (*confirmability*) and are consistent across different contexts and time frames (*dependability*). Credibility and transferability focus more on the quality of the findings within their specific context [48]. Rigorous results accurately reflect the truth value and the participants' perspectives on the phenomenon of interest (*credibility*) and are applicable in a wide range of situations (*transferability*).

Qualitative research can ensure rigor through various methods, including applying thorough data collection methods, data triangulation, researcher triangulation, and reflexivity. While the first three have been introduced earlier in this paper, reflexivity is a term that has not been discussed yet.

Reflexivity can target diverse facets of the research process. In order to address all pertinent dimensions, we categorize it into two distinct forms: personal and epistemological reflexivity. Personal reflexivity addresses the unintended influences of the researcher's values, beliefs, social identity, and experiences on the study [49]. It also acknowledges that researchers are active participants in the study, and their biases can shape the way they perceive and approach the research world. For instance, researchers may be more likely to note down observations that align with their expectations, and overlook or disregard unexpected findings. Being aware of these preconceptions can help researchers identify their unwanted influences on the data and findings.

Epistemological reflexivity on the other hand involves an examination of the potential impact of the study's design on the research outcomes [50]. Epistemological reflexivity enables researchers to reflect on questions such as, "Could alternative methods be employed to investigate the research questions?" or "How does the study's design impact the data and findings?". Engaging in epistemological reflexivity allows researchers to critically evaluate the data's origin and how the study's design has influenced specific findings. Data collection can impact the

direction of research, and epistemological reflexivity can help researchers recognize this and make necessary adjustments.

Saturation is another critical term in qualitative research, referring to the point where new data does not yield any new information [51]. Achieving saturation is often a goal for researchers as it allows them to determine when data collection can stop. To achieve saturation researchers employ theoretical sampling, where participants are selected based on their potential to advance the research [52]. This approach requires analyzing data during data collection to determine the characteristics of subjects that can further the research. Researchers can focus on negative cases, those that differ significantly from the majority of cases, to ensure the study's completeness [53].

At the beginning of a qualitative study, it is often unclear what constitutes sufficient data, that is, enough data to achieve saturation. Also the research focus may shift throughout the study. Therefore, it is recommended to commence data analysis as soon as part of the data is available and continuously reflect on the research process, making adjustments as necessary, as this is inherent in qualitative studies.

GLS employed multiple measures to ensure the rigor and quality of their research. Firstly, they utilized method triangulation by conducting both observations and interviews to increase the credibility and reliability of their findings. Additionally, they employed researcher triangulation during data collection and analysis to reduce the potential for bias, that is, confirmability, and increase objectivity. They also used theoretical embedding to facilitate the interpretation of the results by incorporating sensitizing concepts.

To achieve data saturation within the predetermined fixed time available for data collection, GLS collected as much data as possible. They used theoretical sampling to increase the efficiency of data collection by providing their researchers with specific characteristics to look for, such as non-romantic couples or larger groups walking together, and by paying attention to negative cases. Initially, GLS recorded participants' gender and age during the interviews. However, it soon became evident that these demographic characteristics were not analytically relevant to the aims of the study, as no meaningful patterns related to demographics emerged from the data. Instead, the social contexts in which participants operated proved to be of greater relevance. Consequently, GLS discontinued the systematic documentation of demographic variables and redirected their focus toward aspects more pertinent to the research objectives.

They also incorporated various methods to promote both forms of reflexivity in their study. They held regular meetings with different groups of researchers to discuss the collected data and preliminary analyses. Additionally, they conducted weekly meetings with researchers to monitor the progress of data collection and review the field notes. Through this process, GLS identified that the field notes lacked sufficient detail on certain aspects that the observers took for granted, such as the relationships between people walking together. To address this issue, they instructed the observers to take more detailed factual notes to enrich the data.

Finally, through epistemological reflexivity, GLS recognized that the data collected during the first wave was insufficient to answer their research question thoroughly. Rather than adjusting the research goal, they decided to use the second wave of data collection to gather the additional data required, hence the decision to use semi-structured interviews, with the topic list representing the relevant themes to discuss.

## 2.4 Qualitative data analysis

Throughout data collection, the (initial) analysis of data and constant reflecting and adjusting play a pivotal role. Also within data analysis, the stages presented in this section are subject to

constant reflection and potential going back and forth between them. Nevertheless, we structured this section in three parts, describing subsequent stages, to enhance clarity. In the first subsection, the preparation phase is discussed. Initial choices concern how data are formatted, what software is used, and which analysis framework will be applied. We will emphasis the role and impact of such choices specifically for the context of experimental qualitative data analysis.

Given the choice for a constructivist grounded theory approach, the actual analysis contained the phases of open and axial coding (second subsection) and the final phase of selective coding (last subsection). Selective coding results in the construction of a theoretical model. GLS initially only had data from the first wave and processed and analyzed it up until the selective coding phase. With the realization that information was lacking for the goals at hand and with the opportunity of a second wave of data collection, the final analysis used data from both waves. So, although the second wave data was needed to be able to construct the final theoretical model, it is important to note that all data undergo rigorous processing throughout all stages of data analysis. Illustrations of this process will be provided throughout this section.

**2.4.1 Preparations.** Some considerations and choices need to be made before starting the actual data analysis. Not only are there different analytical approaches and several software packages for qualitative data analysis, there are also different ways to make group comparisons when using qualitative data to evaluate effects in experimental designs. An important aspect in making some of these choices is the outcome that the study at hand aims for, as determined by the type of research question that was formulated. Kearney [54] makes a distinction between descriptive, exploratory, and explanatory research questions. Descriptive and exploratory questions aim for a relatively superficial understanding of aspects or themes at play in the situation being studied and typically result in models that remain close to the data. In contrast, explanatory research aims to provide a more thorough understanding of the process under investigation and often leads to the development of more abstract models, for which the data is transformed to a higher degree [55]. An experiment is to some extent inherently explanatory, since the main goal is to evaluate if an intervention has effect on an outcome. However, the outcomes of interest can be more or less precisely defined, a priori, and this determines whether the study is more exploratory, descriptive, or explanatory.

An often used paradigm in qualitative studies is constructivist grounded theory (CGT) as developed by Charmaz [56]. The CGT approach aims to represent the perspectives of the participants and therefore fits well with behavioral research. The data is processed in a structured, step-by-step approach in such a way that the data speaks for itself. Some qualitative researchers argue that the high level of structure of CGT limits the creative space characteristic of qualitative research [57]. Other approaches, such as discourse analysis, narrative analysis, and phenomenology, are less structured than CGT and can be more interpretative. Discourse analysis seeks to explain behavior from an outsider perspective [58], narrative analysis focuses on reconstructing complete stories [59], and phenomenology aims to describe experiences [60]. Depending on the context of the research each of these approaches could be used for the evaluation of experiments.

The treatment or intervention of interest creates groups in the data. In a qualitative analysis of an intervention effect, one can incorporate this grouping in two ways. First, all data can be analyzed as if coming from one population (e.g. [20,61]). In this case, the last phase of the analysis is comparing the resulting model between the groups. The description of a potential intervention effect is in terms of certain parts of the model originating from one group and not the other, or in terms of certain relations being more prominent in one of the groups. This approach may be the preferred choice when the research question is merely descriptive or

exploratory; it may reveal unexpected findings, such as themes that are entirely missing from one subgroup or a process that is not part of the other. Alternatively, separate analyses can be done for the treatment conditions (e.g. [17]). Then distinct models are built for the data from each of the groups and the final phase consists of comparing the resulting models. This approach is suitable when it is assumed that groups differ inherently and independent processes are expected to be evaluated. This approach seems more natural for studies where aims and questions are of a more explanatory nature.

Irrespective of the analytical approach and the way the grouping structure is incorporated, appropriate software is required to systematically analyze the data. Various computer-assisted qualitative data analysis software (CAQDAS) programs are available and assist researchers in structuring their analyses by identifying (sub)groups in the data, structuring codes, creating separate models, and running queries. Well-known CAQDAS programs include MAXQDA [62], Atlas.ti [63], and NVivo [64], among others. All these programs provide functionalities such as coding, compatibility with multiple data types, specification of demographics per document or case, memo writing and linking, query usage, and project file sharing. Researchers can choose a CAQDAS based on their specific needs, familiarity with the software, or availability.

GLS used CGT to answer the explanatory research question on how the empathy-inducing nudges of the intervention program potentially influenced the decision making process with respect to adherence to the distance regulations. NVivo Release 1 [64] was used for the data analysis, since it satisfied their criteria and was readily available through a campus license. Within NVivo, they distinguished between the control and intervention groups by creating attributes for each case, including control or experimental group, day, time, interviewer/observer, and week.

Despite the explanatory nature of the research question, GLS assumed that similar processes would be present in the control and experimental conditions. They therefore choose to analyse all data together to create one overall model. However, the intervention could lead to different weights in parts of this model, for instance, certain concepts or relations may be only present in one of the groups, while other parts of the model may be more prominent in one group than in the other. If such group differences emerge, GLS can conclude that the intervention imposing empathy-inducing nudges triggered different processes and thus was effective.

**2.4.2 Open and axial coding.**   In the remainder of this section we will describe the actual data analysis given the choice for a CGT approach. In the grounded theory tradition, three phases in the coding of data are subsequently conducted: open coding, axial coding and selective coding [33,65].

During open coding, the aim is to identify and list all topics presented in the data by allowing to let the data speak for itself. Preconceived notions may interfere with the analysis and, therefore, it is important to avoid imposing structure at this point. To ensure the quality of open codes, researchers should keep an open mind and be prepared for unexpected themes to emerge. However, to fully understand the phenomenon of interest, it can also be beneficial to incorporate the earlier identified sensitizing concepts in this phase of coding. Although the sensitizing concepts are typically too broad to include directly, they can be translated into so-called a priori codes [66]. A priori codes are more specific then sensitizing concepts, and provide a link between data and theory. The use of a priori codes enables identification of the specific ways in which sensitizing concepts might occur in the data and helps interpret the data. An example of a sensitizing concept could be *empathy*, which is a theoretical, abstract construct. Participants will most likely not mention or express this construct directly, so a priori codes are needed. Examples of a priori codes for *empathy* could be 'protecting a loved one'

(for interview data), or 'steps back so other people can join a conversation' (for observational data).

Effective open coding involves adhering to the 3C's: content, context, and coverage. This entails applying labels that describe the content of a fragment, which spans enough data to capture the context, while ensuring that the created codes cover all topics in the data [67]. The open coding phase continues until saturation is reached, that is, until adding new data does not lead to new open codes [68].

The next phase is axial coding, which aims to structure the data and identify latent themes by exploring the relationships between the open codes that emerged from the data [69]. Although sensitizing concepts can be used to provide an initial structure, it is important to remain open-minded and avoid tunnel vision. By exploring different possible structures, researchers can gain a more nuanced understanding of the various layers in the data and potentially uncover new perspectives that were not previously considered. Throughout this process, the method of constant comparison is applied to ensure that themes are consistent across the data [70]. This involves checking how themes are present in the data of other participants and how they may differ in terms of their characteristics.

Axial coding results in a structured code tree, with well-defined main codes (themes), and relevant sub-themes and open codes placed underneath the overarching themes. Before discussing the next phase in the data analysis, we will first continue with a description of the initial analyses in GLS, and present some additional considerations that are important before moving to the third phase of analysis: selective coding.

In GLS, the open coding of the data from the first wave was conducted by six researchers ensuring researcher triangulation. Although GLS considered creating a priori codes for the sensitizing concept *empathy*, they decided against it to avoid over-fitting the data and over-interpreting the findings. Codes were created to describe the content, selecting fragments to capture their context, and cover all themes in the data as prescribed by the 3C's approach. For instance, they produced codes such as *Waits for passersby to pass*, *Avoid putting entire house in quarantine*, *Keeping your distance with friend is weird*, and *Avoids grandparents due to health issues*. Saturation was achieved after coding approximately 60% of the observations and 80% of the short interviews.

The axial coding phase was carried out by two of the six researchers doing the open coding, and two additional researchers. In this phase the sensitizing concept *empathy* was applied as one of the themes, since they wanted to investigate its impact on adhering to COVID-19-related measures. Data from the observations served well for the identification of behavior and behavior changes. The data from the short interviews was supposed to add insights into participants' strategies and motivations, but in this phase of analyzing the data it became clear that the short interviews provided insufficient information to fully address their research goals. With the current data GLS could only provide a description of visible behavior, the direct influence of the nudges, and a very general enumeration of considerations for the behavior. It became clear that an in-depth understanding to explain which motivations were active in both groups, and how that led to certain strategies to behave in relation to COVID-19-measures, would not be achieved.

In such a situation there are two options: collecting additional data (if possible), or reporting the insights gained from the available data, even if the original questions cannot be fully answered. GLS were fortunate that a second wave of data collection was announced, allowing them to collect more in-depth data and to pursue a complete answer to their original research question.

Resuming the data analysis with the additional data from the interviews, in the axial coding phase they initially still used the overarching theme of *empathy* to view the data

through this lens. With the data from the second wave however, at some point they decided to drop *empathy* as a sensitizing concept. It appeared to lead to over-fitting the data, since most themes could be interpreted as sub-themes of *empathy*. Excluding *empathy* enabled the researchers to gain a more comprehensive understanding of the data.

The new data provided more context and explanation to the observations' results and supplemented the information acquired from the short interviews. To address the research goal, two main themes were formulated: *Motives* and *Strategies*. Several sub-themes were identified that represented aspects of these main themes, e.g., 'no fear' under motives, or 'use of self-tests' under strategies. A further illustration of part of the coding scheme is provided in Table 1 and shows how themes and underlying sub-themes emerged from observations and text fragments from the interviews. Such a visualization of results forms the basis for the final phase, that is, selective coding.

**2.4.3 Selective coding.** Selective coding is the third and final phase of data analysis in the grounded theory tradition [71]. This phase is the culmination of the study process, which allows the qualitative researcher to engage in a dialogue with the data to search for the most accurate representation of the truth value. Its main objective is to develop a theoretical model that explains the phenomenon under investigation.

The outcome of axial coding, which includes the code tree, definitions, and memos generated during the open and axial coding phases, serves as the foundation for building the theoretical model in the selective coding phase. The selective coding process involves identifying the relationships between the themes that emerged from the previous phases, such as developing a chronological story, typology, or identifying which themes can or cannot coexist [13]. Providing clear instructions for this phase can be challenging, as the approach may vary depending on the research situation. A common strategy is to identify a core category

**Table 1. A selection of subthemes, open codes, and fragments to serve as illustrations for the coding scheme resulting from the axial coding phase.**

| Theme | Subtheme | Open code | Fragment |
|---|---|---|---|
| Motives | Influence others | Mother sees little risk in being close | 'Well my mother, she works at home. And she only goes to the supermarket and back. There is so little risk of being infected.' 'That is how I rationalize it: she is vaccinated, she sees barely any people, I don't see many either. We think it is okay.' |
| | | Keeping friends at a distance feels weird | 'It is so weird to keep your distance if you are not used to doing that. That is why we don't do that.' 'If I wanted to [edit: keep distance between friends], it is not realistic.' |
| | | "Hugger" kept giving hugs | 'One friend was always used to giving hugs. And he kept doing that. But with other friends we give a box.' |
| | No fear for COVID-19 | Risk of getting infected is low | 'For as far as I can tell, the risk for us of getting infected is low.' 'My friends, the four I regularly see, are very calm, timid, boys. They don't go out or do crazy student things. So if you are at home, the chance is much lower of getting infected. I take that risk.' |
| | | Risk of infection is low | 'I am still young, I am not afraid I will get really sick from the virus.' |
| Strategies | Use of self-tests | Tests regularly | 'I meet a small number of different people, most of whom are vaccinated by now, and I test regularly, because I received a self-test-thingy at the University.' |
| | | Testing before meeting family | 'Before going home for Christmas I did a self-test to be sure that I had not caught COVID-19, so I could go home without too much risk.' |
| | Limited number of people gathering | Mostly one-on-one meetings | 'I meet friends mostly one on one.' 'We had the rule to only allow one external person in the kitchen at any time.' |
| | | Avoiding large gatherings | 'We are relatively safe, we do not organize or attend any large illegal parties.' 'Well I invite less people to my home.' |

that connects the themes previously identified [70]. This strategy is particularly useful when multiple models are created and each sub-population needs to be analyzed, as is typically the case in experimental designs. The core category acts as the missing piece that connects the different models and explains differences between the sub-populations. This will be further elaborated in the subsection about evaluating the intervention effect.

Throughout selective coding the process of *constant comparison* is applied [13]. With constant comparison researchers continuously check how modifications to the model work for other, already analyzed, data. A potential pitfall to qualitative research is to let the final round of data collection be the deciding factor for the final model, that is, the patterns present in the last observed data are used to determine the definitive descriptions. Through constant comparison, each addition or adaptation of the model is verified with earlier data. This process is essential to ensure that the final model correctly represents the participants' views.

The outcome of selective coding is a theoretical model that explains how participants perceive the phenomenon or process of interest [55]. The resulting theoretical model should adhere to four key principles: they should *fit* within the field of study, be *understandable* to those in the field, be *generalizable*, and provide some level of *control* to the researcher in the form of understanding the process or phenomenon of interest [68]. By setting up the study as described in the previous sections, the results acquired should meet these criteria. The use of sensitizing concepts helps fit the theory within the existing literature, while theoretical sampling ensures generalizability. The CGT coding procedure ensures that the resulting theory is understandable and provides control to the researcher.

To illustrate the process of creating and verifying a theoretical model from the qualitative data, we present the results of GLS in Fig 2. Here, we zoom in on a few aspects of the model, to give an impression of the process and result of selective coding. Note that in this phase they did not distinguish between participants in the intervention or control condition; they built one overall model for all data obtained.

In GLS, the stories of participants demonstrated some common ground that helped understand different aspects of their behavior and motivations. For instance, the researchers observed a recurring pattern in the participants' explanation of circumstances that affected their choices for actual behavior. From observations and interviews it became clear that social interactions are context-dependent, that is, behavior does not only depend on intentions (based on reasoning about measures and the motivations to adhere to them) but also on, for instance, how friends or other contacts behave, what their intentions are, and on perceived risk of not behaving in the intended way (see Fig 2). To give just one illustration, the following quote is exemplary for several similar remarks about the influence of other people's behavior:

Participant D2505B2: '*It is sort of legitimized because the people around you don't do it either.*'

To verify the conclusion that social interactions are context-dependent, a revisit of all data was made to validate this pattern with the data itself as the method of constant comparison describes. An example of such a verification in the data was given by a participant that described intended behavior depending on who they were with.

Participant D1006O1: '*When I'm with family, then I'm a bit easier about it [edit: 'it' refers to keeping distance]. But if I'm just in the supermarket then I'm a bit more careful with that. So that's actually kind of where I draw the line. People close to me, that's where I am very lenient with it. But if it's just about public places then I am more cautious.*'

The researchers saw similar things in the observations happening. Most people ignored the station for hand sanitation, but the following happened when people entered together:

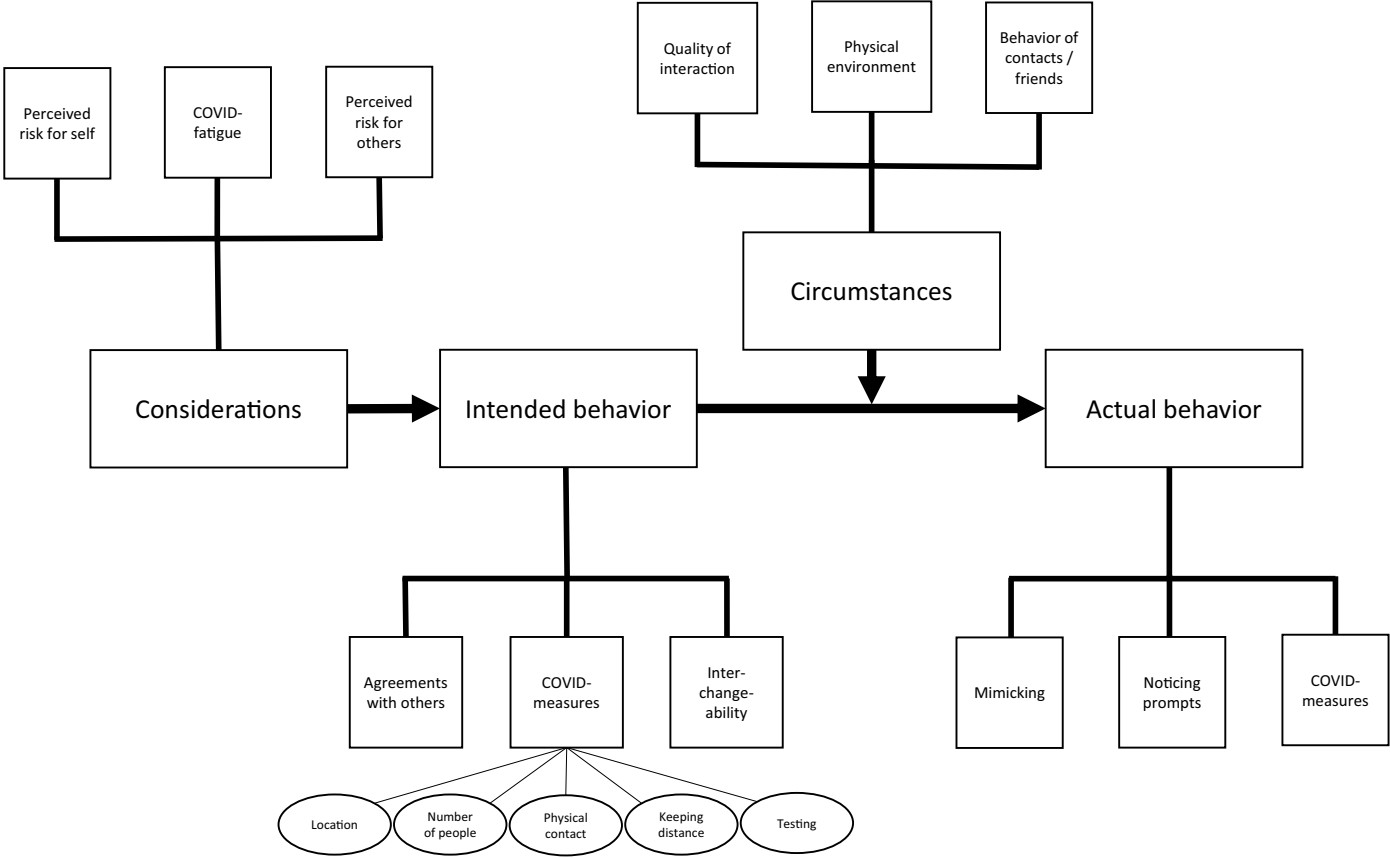

**Fig 2. Theoretical model as a result from the selective coding, where forms of behavior are explained by different kinds of circumstances, intended behavior, and considerations.**

Observation M1611D215: '*PP does not keep distance from either friends. First friend disinfects their hands, they stand close to each other while waiting. When the first friend is done, PP disinfects their hands, and waits until the third person has disinfected their hands as well. When they walk away they do not keep distance.*'

Similar things happened on a regular basis: when one person of a group adheres to a measure, the others follow their lead. With the developed theoretical model, GLS could more easily classify new data and categorize topics mentioned by the participants.

## 2.5 The intervention effect

The final phase in determining evidence for a potential intervention effect is the comparison of the intervention and control group. In research where separate models are created for the groups, this means comparing the models on their similarities and differences. This entails describing the processes per group, in order to understand at what point they take meaningfully different paths. Subsequently, the data is again consulted to get a comprehensive understanding of participant's perspectives in both groups. The final challenge is to describe them accordingly, so the readers can grasp the essential characteristics for both groups.

In research where a singular model is employed, researchers often utilize queries as valuable tools provided by CAQDAS. These queries generate cross-tabulations, which elucidate the frequency of various aspects of the theoretical framework observed within different groups. A basic query involves tallying the occurrences of specific (sub)themes across these groups. Furthermore, researchers can analyze the co-occurrence of themes and discern potential disparities between intervention and control groups regarding the prevalence of theme relationships. However, it is imperative to note that while queries offer initial insights into the underlying processes, a thorough understanding and interpretation of the results within a theoretical framework necessitate continual data analysis through constant comparison.

In addition to quantifying the results with queries, a qualitative assessment is indispensable. Qualitative researchers often describe this process as "engaging in a dialogue with the data". Essentially, researchers aim to uncover potential relationships using various tools such as queries, memos, and theoretical frameworks. These relationships are then rigorously tested by revisiting the original raw data, a practice akin to the concept of constant comparison introduced earlier.

GLS categorized all data entries as belonging to either the intervention or control group. Table 2 provides an illustration of some comparison between these groups. The researchers chose the themes to inspect partly based on the theory behind how prompting with empathy would influence behavior: specifically the subthemes under *motivation*. The table displays the proportions of participants from each group who reported a particular theme or combination of themes. By examining these proportions, GLS could gain initial insights into the potential differences between the groups and identify the particular aspects in which they may diverge.

Subsequently, the investigated queries prompted the generation of preliminary explanations, which in the context of abductive reasoning [11] can be seen as a form of (data-generated) hypotheses. To illustrate the analytical process, let's consider the theme of *No fear for COVID-19*, which was mentioned less frequently in the control group compared to the intervention group (control: .16; intervention: .50). The researchers' initial expectation was that this difference might have resulted in divergent intentions to adhere to COVID-19-related measures. However, upon revisiting the data, no clear differences in intentions between the groups were observed, that is, the intervention group did not overall seem less willing to adhere to COVID-19-measures than the control group.

**Table 2. Example of query used to process selective coding to crosstabs: proportion of participants mentioning subthemes of strategy, motivation, and relations between subthemes in the control and intervention groups.**

| Used strategy | Control | Intervention |
|---|---|---|
| Limited number of people | .64 | .33 |
| Meeting outdoors | .24 | .33 |
| Use of self-tests | .15 | .50 |
| Behavior contacts/friends | .64 | .67 |
| **Motivation** | **Control** | **Intervention** |
| No fear for COVID-19 | .16 | .50 |
| Perceived risk for self | .64 | .67 |
| Duration of measurements | .36 | .67 |
| **Relations** | **Control** | **Intervention** |
| Perceived risk for self vs. Use of self-tests | .12 | .33 |
| Perceived risk for self vs. Meeting outdoors | .12 | .00 |

To explain the finding that the intervention group, despite mentioning the motivation *No fear for COVID-19* more frequently, expressed similar intentions to adhere to COVID-19-related measures, a new preliminary explanation was formulated. GLS speculated that differences in *empathy*, which the intervention aimed to induce, could account for the absence of divergent intentions. It was theorized that although fewer participants experienced fear for COVID-19 themselves, the empathy-based nudges might have influenced their intended behavior, resulting in a comparable level of adherence to measures. However, upon conducting another reassessment of the data, this preliminary explanation was not supported either. Instead, during this round of data reassessment, two other topics consistently emerged in conjunction with the mention of *No fear for COVID-19*. It became apparent that participants who expressed no fear for COVID-19 often mentioned employing specific strategies to mitigate the risk, such as using self-tests and limiting their social interactions to a small number of people. The following quote serves as an illustration of this observation:

Participant E1905L1: '*I have one group of friends that I actually see [edit: without keeping distance] every week. Yes, if they all stay within that group of friends, then it's not too bad with how many contacts you have*'.

The researchers' final theory was that, notwithstanding minor differences between the groups, no relevant effects attributable to the prompts were observable. To validate this assertion, they conducted a test wherein two researchers were tasked with reviewing two transcripts of semi-structured interviews, while the experimental status was masked, and asked to allocate these to either the control or experimental group. The researchers were unable to accurately allocate the transcripts, supporting the preliminary explanation that the prompts had no relevant effects in this study. A further discussion of the empirical findings of the COVID-19-study is beyond the scope of this paper and can be found in Glebbeek et al. [25].

With this the data analysis is concluded. Our intention was to demonstrate the iterative process of preliminary explanation generation and validation through repeated data reassessment. This iterative approach is vital in answering the central question of qualitative evaluation: how can we describe and comprehend the effects of the intervention on the desired outcomes? By engaging in this meticulous and iterative process, researchers strive to gain a comprehensive understanding of the intervention's impact on participants' behavior and the motivation for this behavior. In the context of the quantitative [24] and qualitative [25] studies referred to in this paper, in both studies it is concluded that no support for an intervention effect was found. However, by developing a model based on empirical qualitative data, GLS were able to provide deeper insight into the underlying mechanisms behind the quantitative findings, offering an explanation of the internal processes that contributed to those outcomes.

## 3 Discussion

In this methodological tutorial, we provided guidance to assist (quantitative) researchers in incorporating qualitative methods into their experiments. Compared to existing literature in this area, we aimed to provide a practical and comprehensive overview of the stages of the research process from start to finish. Furthermore, the focus on the application of qualitative research to experimental designs is a contribution in the field of behavioral sciences that was missing until now. We argued that, by following the stages outlined in Fig 1, researchers

can obtain a more holistic understanding of their research problem than with the use of solely quantitative methods. Here, we add Table 3, which summarizes the essential elements for conducting a rigorous qualitative investigation of potential intervention effects in experimental designs, and gives an overview of some key references for these elements. With this paper we aimed to guide researchers who are well-versed in quantitative methods towards the integration of qualitative methods into their study. By providing detailed guidance and an illustration, we hope to facilitate a smooth transition and ensure the quality and rigor of qualitative experiments.

Previous literature on qualitative experiments in behavioral sciences primarily engaged in philosophical contemplation regarding the suitability of qualitative methods within experimental contexts [16,18], or provided individual empirical reports [19,20,e.g., ]. In contrast, our focus is on highlighting the necessary stages that researchers must undertake to conduct their own qualitative evaluation of an experimental intervention. Where earlier studies focused mainly on the design of the research [17], we discuss considerations and decisions for all stages of the project. Notably, our contribution places a greater emphasis on the application of reflexivity throughout the study and advocates for a flexible and iterative process.

The degree of flexibility inherent to qualitative research is one of the main differences compared to the quantitative approach. Refining the study design throughout the research process adds to the rigor of the research. In quantitative research, there should be a clear a priori plan for data collection and analysis. If it is needed to explore how different elements work, a pilot study can be employed before the actual study, and these pilot data should not be included

**Table 3. The most relevant topics to consider when evaluating experiments with qualitative methods, ordered per stage of the research.**

| Stage | Short description of stage | Term | References |
|---|---|---|---|
| Research goal | In this stage the basis for the research is layed down. The approach is to be determined and the scope of the research is formulated, making use of the strengths of qualitative methods to add value. | Scope creep | [31] |
| | | Holistic approach | [8,10] |
| | | Sensitizing concepts | [13] |
| | | Constructivist GT | [56] |
| | | Discourse analysis | [58] |
| | | Narrative analysis | [59] |
| | | Phenomenology | [60] |
| | | Content analysis | [72] |
| Data collection | During this stage, regardless of the approach, the data collection is designed, making sure the data is collected such that it leads to a comprehensive answer to the research question. | Qualitative data | [35] |
| | | Observations | [36,37] |
| | | Researcher triangulation | [44,45] |
| | | Data & method triangulation | [43] |
| | | Interviews | [39,40] |
| | | Topic list | [41,42] |
| | | Verbatim transcription | [46] |
| Reflecting & Adjusting | It is important to critically assess the rigor of the study. By applying multiple strategies to improve different aspects of rigor, the quality of the research can be held high. | Rigor | [43,47] |
| | | Reflexivity (epistemological) | [50] |
| | | Reflexivity (personal) | [49] |
| | | Saturation | [51] |
| | | Theoretical sampling | [52] |
| | | Negative cases | [53] |
| Data analysis | As soon as the first data is collected the analysis will start. Essential in the approach of constructivist grounded theory is to keep an open mind for unexpected information. | Open coding | [71] |
| | | A priori codes | [66] |
| | | Axial coding | [69,71] |
| | | Selective coding | [71] |
| | | Core category | [70] |
| | | Constant comparison | [13] |
| Intervention effect | In the final research stage, the comparison between experimental groups is conducted to determine the intervention's effect, if any. | Types of theoretical models | [55] |
| | | Key principles of theories | [68] |

in the actual analysis. In contrast, in qualitative research adjustments to the study design and analysis can be made as needed throughout the study. Reflexivity should improve the rigor of the study and assist in finding the most reliable and comprehensive answer possible. In the worked example, GLS adjusted their plans several times; e.g., in the second wave of data collection they included more extensive interviews to get a better understanding of the motives and strategies of the participants, and in different phases of the analysis they moved back and forth between ignoring or enforcing *empathy* as a sensitizing concept or through a priori codes, to search for the right balance between guiding and structuring the data analysis and letting the data speak for itself without giving too much direction.

To elaborate a bit on this last point, the approach to data analysis in general and the sensitizing concept of *empathy*, specifically, was highly flexible. While theory suggests that a priori codes should be formulated for sensitizing concepts, the researchers chose not to do so, as they were concerned that this could lead to over-fitting the data. At the start of axial coding, they explored the fit of the sensitizing concept with the data, a common practice in qualitative research. Ultimately, the researchers made the decision to discard the concept again at a certain point during axial coding. This shows once more that qualitative studies involve ongoing reflection, iteration, and adaptation: even in a constructivist grounded theory approach, known to be relatively structured compared to other qualitative approaches. The decision-making process should be driven by the research goals and commitment to rigor and transparency.

Two limitations of our study are important to mention. In the second wave of data collection in the empirical example, GLS managed to interview only a small number of participants in the intervention group. Even when invited for the interview near the intervention location, most people preferred the offered option of doing the interview at another time, often incidentally during a control week, or they preferred an online interview. This changed their status to control group participants since they did not encounter the intervention shortly before the interview. Despite observing this throughout the data collection phase, and stimulating the interviewers to motivate people to be interviewed at the intervention location during an intervention week, the final number remained low. It is a clear example of reflecting and adjusting throughout the study (by adapting instructions to interviewers), but in this example it did not lead to more interviews in the experimental condition. We believe that negative impact was manageable and that the data was still rich enough to draw conclusions with regards to the absence of an effect of the interventions. Either way, the worked example did enable a thorough testing of all methodological stages, including the final comparison between intervention and control group, leading to the methodological recommendations presented in this paper. Though a more exhausting experimental group might have enriched the data analysis even further, we think the steps identified would not have changed with more data.

Basing our paper largely on the worked example, however, has another limitation and that is the somewhat narrow look on the different opportunities that qualitative research methods offer. We discussed several approaches in relative depth (e.g., interviews, observation, the CGT approach) because they were applied in the worked example. And we left out others (e.g., focus groups, archival data, case studies) that were not applied in the worked example. In order to be more exhaustive, in Table 3 several additional approaches and methods are listed, including relevant literature references.

Following our recommendations, supplemented by the information in Table 3, researchers will gain the necessary tools to effectively employ qualitative methodologies for evaluating experimental effects. Compared to our example, it will be more straightforward to apply this

approach in more controlled experimental settings, like in a laboratory environment. An evident challenge highlighted in the worked example was the variance in participant engagement with the intervention, a common occurrence in field experiments where researchers contend with less control compared to lab experiments. Augmenting or even replacing traditional quantitative methods with qualitative approaches could lead to unforeseen revelations regarding intervention effects.

In conclusion, our paper provides guidelines for the qualitative evaluation of experimental designs. While these guidelines are not all-encompassing, they provide a useful starting point for researchers looking to incorporate qualitative methods in their experimental research. We hope to have convinced readers that sometimes experiments in social and behavioral research can be studied in a more meaningful and more insightful way if one either changes to qualitative methods, or adds a qualitative evaluation of results to the more traditional quantitative way of measuring intervention effects.

## Acknowledgments

We are grateful to Prof. Dr. Denise de Ridder for setting up the bigger context study and allowing for the qualitative study that served as a worked example in this study to be executed. We thank Dr. Marie-Louise Glebbeek for offering a critical reflection during the design and execution of the empirical study. We thank Efe Adu, Timo Keijzer, Jamie Keurntjes, Jonas van Oosten, & Danja Roelofs for their contributions to the data collection. We used ChatGPT for textual improvements.

The preregistration for this study can be accessed via this link: https://archive.org/details/osf-registrations-knb2d-v1

## Supporting information

**Appendix A. Observation list control week.** The translated version of the observation list as was used during the control weeks.
(PDF)

**Appendix B. Observation list intervention weeks.** The translated version of the observation list as was used during the intervention weeks.
(PDF)

**Appendix C. Topic list short interviews.** The translated version of the topic list as was used during the short interviews.
(PDF)

**Appendix D. Topic list long interviews.** The translated version of the topic list as was used during the long semi-structured interviews.
(PDF)

## Author contributions

**Conceptualization:** Hidde Jelmer Leplaa, Irene Klugkist.

**Data curation:** Hidde Jelmer Leplaa, Mariska Bouterse, Karlijn F.B. Soppe.

**Formal analysis:** Hidde Jelmer Leplaa, Jari A. Tönjes, Mariska Bouterse, Karlijn F.B. Soppe, Irene Klugkist.

**Investigation:** Hidde Jelmer Leplaa.

**Methodology:** Hidde Jelmer Leplaa, Karlijn F.B. Soppe, Irene Klugkist.

**Supervision:** Irene Klugkist.

**Validation:** Hidde Jelmer Leplaa, Karlijn F.B. Soppe, Irene Klugkist.

**Visualization:** Hidde Jelmer Leplaa, Irene Klugkist.

**Writing – original draft:** Hidde Jelmer Leplaa, Jari A. Tönjes, Mariska Bouterse, Karlijn F.B. Soppe, Irene Klugkist.

**Writing – review & editing:** Hidde Jelmer Leplaa, Karlijn F.B. Soppe, Irene Klugkist.

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
