## [Decision Letter · Decision Letter 0]

11 Oct 2024

PONE-D-24-29638A qualitative evaluation of an experiment: Studying the effects of empathy-inducing probes on distancing during COVID-19 to derive methodological guidelines.PLOS ONE

Dear Dr. Leplaa,

Thank you for submitting your manuscript to PLOS ONE. After careful consideration, we feel that it has merit but does not fully meet PLOS ONE’s publication criteria as it currently stands. Therefore, we invite you to submit a revised version of the manuscript that addresses the points raised during the review process. This manuscript has a lot of value and this approach in particular enhances the understanding of quantitative results through qualitative findings. At the same time, both reviewers bring up serious concerns that I share. I will note that my decision was on the border between major revision and reject, but I do see the value of this manuscript and want to give the opportunity to address these concerns. In particular, I would note:

Need to qualify language and assertions (mentioned by both reviewers)Clarify purposes of the analyses (see Reviewer 1)Specifically address point 7 from Reviewer 1

We look forward to receiving your revised manuscript.

Kind regards,

Gabriel Velez, Ph.D.

Academic Editor

PLOS ONE

Journal Requirements: When submitting your revision, we need you to address these additional requirements. 1. Please ensure that your manuscript meets PLOS ONE's style requirements, including those for file naming. The PLOS ONE style templates can be found at https://journals.plos.org/plosone/s/file?id=wjVg/PLOSOne_formatting_sample_main_body.pdf and https://journals.plos.org/plosone/s/file?id=ba62/PLOSOne_formatting_sample_title_authors_affiliations.pdf 2. Please update your submission to use the PLOS LaTeX template. The template and more information on our requirements for LaTeX submissions can be found at http://journals.plos.org/plosone/s/latex. 3. In the online submission form, you indicated that "We did not collect data ourselves for this study. In this methodological paper we propose a specific way to analyze experimental research. Data was obtained through contact with the authors of an empirical study, which will be submitted in the next few months. We have an overlap with some of the authors, which made permission to support our methodological approach with a proof of conduct realistic. The authors of the other paper are the owners of the data, and decided to make it available upon request. In their paper they will specify the necessary procedure to acquire the data." All PLOS journals now require all data underlying the findings described in their manuscript to be freely available to other researchers, either 1. In a public repository, 2. Within the manuscript itself, or 3. Uploaded as supplementary information.This policy applies to all data except where public deposition would breach compliance with the protocol approved by your research ethics board. If your data cannot be made publicly available for ethical or legal reasons (e.g., public availability would compromise patient privacy), please explain your reasons on resubmission and your exemption request will be escalated for approval. 4. Please include your full ethics statement in the ‘Methods’ section of your manuscript file. In your statement, please include the full name of the IRB or ethics committee who approved or waived your study, as well as whether or not you obtained informed written or verbal consent. If consent was waived for your study, please include this information in your statement as well. 5. Please include a separate caption for each figure in your manuscript.

Reviewers' comments:

Reviewer's Responses to Questions

**Comments to the Author**

1. Is the manuscript technically sound, and do the data support the conclusions?

Reviewer #1: No

Reviewer #2: Yes

2. Has the statistical analysis been performed appropriately and rigorously? 

Reviewer #1: No

Reviewer #2: Yes

3. Have the authors made all data underlying the findings in their manuscript fully available?

Reviewer #1: No

Reviewer #2: Yes

4. Is the manuscript presented in an intelligible fashion and written in standard English?

Reviewer #1: Yes

Reviewer #2: Yes

5. Review Comments to the Author

Reviewer #1: Dear authors,

The article submitted for my review is entitled “A qualitative evaluation of an experiment: Studying the effects of empathy-inducing probes on distancing during COVID-19 to derive methodological guidelines”. The article is classically structured and clearly argued. The theme of the article is stimulating: designing a qualitative evaluation method for a quasi-experiment. Nevertheless, I feel that certain elements of the article need to be improved or supplemented. I'd like to point them out below:

1) In the introduction, despite the authors' didactic efforts, the distinction between experience and quasi-experience is not clear. The authors do not give a clear definition of quasi-experience and what distinguishes it from experience. This is important, however, as the originality of the article seems to rest on the study of a quasi-experience.

2) Still in the introduction, the authors write “there is little literature on using a qualitative approach for the evaluation of experimental designs in behavioral science.” Firstly, I don't really agree with this peremptory assertion. Numerous books or articles deal to a greater or lesser extent with the qualitative evaluation of experimental design in behavioral science (for example: Kite & Whitley, 2012. Principles of research in behavioral science; Orcher, 2016. Conducting research: Social and behavioral science methods; Maxwell 2008. Designing a qualitative study The SAGE handbook of applied social research methods; etc.). Secondly, there are a multitude of methods for evaluating qualitative experiments (Rossman & Wilson, 1985, Numbers and words: Combining quantitative and qualitative methods in a single large-scale evaluation study. Evaluation review, 9(5), 627-643; Horsburgh, 2003, Evaluation of qualitative research. Journal of Clinical Nursing (Wiley-Blackwell), 12(2); Hennink, et al (2020). Qualitative research methods. Sage;Yadav, 2022. Criteria for good qualitative research: A comprehensive review. The Asia-Pacific Education Researcher, 31(6), 679-689. etc.). I fail to see where the contribution of the new method proposed by this article lies. It should be clarified in a very precise and documented way. In particular, it should be specified whether the proposed method concerns quasi-experiments or experiments, whether it concerns mixed or totally qualitative methods, or even experiments that are mainly quantitative but incorporate qualitative data; otherwise, it's hard to see where the article's contribution lies. At the end of the introduction, the authors state “we provide a step-by-step approach step-by-step approach for future researchers wishing to carry out a qualitative evaluation of a"quasi-)experimental design”, but it is not clear whether this is really a research gap. In this respect, in the “Methodological approach” section, the authors state “Our own study is an example of adding a qualitative element to a quantitative study”, so we'd be tempted to believe that the gap lies here, but we'd have to justify it much earlier in the article.

3) Still in this “Methodological approach” section, the authors mention a large number of bibliographical references (references 20 to 61). However, in a research article, bibliographical references usually appear in the introduction or discussion, but not in the methodological or results section. This gives the impression that your work is more like a literature review than a concrete experiment. The authors should clearly state their position on this point. Is the article a methodological review or the testing of a qualitative experimental method? This is not very clear.

4) In the “data collection” section, you state that you have carried out interviews and observations, but it's hard to understand why you are doing so (for what purpose) and with whom (list of interviewees). What's more, if I understand correctly, the purpose of these observations and interviews is to justify the results of the method you describe, but your results contain very little verbatim information about these observations and interviews. So I don't see how these observations and interviews validate your method.

5) Still in the presentation of the method, you insert a few verbatims after figure 2 and then announce that “Observations and interviews clearly showed that social interactions are context-dependent, i.e. that behavior depends not only on intentions (based on reasoning about actions and motivations for complying with them) but also, for example, on how friends or other contacts behave, their intentions and the perceived risk of not behaving as intended (see figure 2).”. It's possible to agree with this statement, but I don't see how it validates your method of evaluating a qualitative experiment? In my opinion, to validate a method, it is necessary to provide very factual evidence, for example by comparing the responses of the experimental group and the control group, and by showing that each stage of the method improves the results. In my opinion, your article fails to do this. Table 2 compares the results between the two groups, but I fail to see how this validates your method. To validate your method, it would probably have been more appropriate to set up two groups, an experimental group studied using the new method, and a control group not studied using this method.

6) To conclude the presentation and justification of your method, you write that “Our intention was to demonstrate the iterative process of hypothesis generation and validation through repeated re-evaluation of the data.

of data.” Firstly, I don't think this intention corresponds to the initial objective of your study set out in the introduction. Secondly, I discover that your method ultimately consists in determining hypotheses by re-evaluating data. Figure 1, which presents your method, and Table 3 do not mention the construction of these hypotheses, which, in my opinion, is part of a qualitative abductive research methodology not presented in this work.

7) I'm not very convinced by your discussion, as you fail to demonstrate the validity of your method. You write “Table 3 summarizes the essential elements for conducting a rigorous qualitative investigation of potential research projects.” Table 3 presents a literature review of the various stages (Research objective; Data collection

Personally, I regularly conduct qualitative studies and apply these steps from the literature, but I've never tested the validity of the overall method, and I don't know if the method is valid. After reading your work, I still don't know. You have carried out a literature review of the design stages of a qualitative study, which is interesting, but it is not possible to say, as you do, that this method is rigorous and reproducible. To write that, it would have been necessary, at the very least, to apply this method to several studies, or to analyze the results between an experimental group and a control group. I'm not saying that your method isn't correct, but you don't provide proof of it.

In conclusion, I'm not convinced by your work and the evidence you provide. You do a good job of analyzing the literature on the stages of qualitative research (but this has already been done several times in the social sciences), but you don't provide any proof of the reliability of your methodological approach. To publish your work, I think you need to start again from Table 3 and test this method in its entirety on several qualitative studies in order to conclude whether or not this method is reliable and rigorous.

Thank you for your attention, and I encourage you to take this work further.

Yours faithfully

Reviewer #2: Dear Editor, Dear Authors,

Thank you for the opportunity to read this interesting manuscript, in which the methodology of a qualitative study is described clearly and step-by-step. In my opinion, qualitative research often holds an unjustly subordinate position in the research landscape compared to quantitative research. However, the qualitative approach provides a valuable opportunity to explain interventions or quantitative results, thereby offering a more comprehensive understanding of (complex) interventions. This is precisely the approach taken by the authors of this manuscript, as they aim to demonstrate a possible implementation and pave the way for future researchers to adopt a similar methodology. However, there are a few points I would like to emphasize. Below, you can find my suggestions:

• Abstract: Please include the qualitative research design, specifying that the qualitative evaluation was conducted using Grounded Theory. I would also suggest clarifying this earlier in the manuscript to help guide the reader (e.g., a guide for a qualitative evaluation based on Grounded Theory).

• In your manuscript, you frequently refer to the 'effect of the intervention' or 'intervention effect' in the context of the qualitative approach. The term 'effect' is more commonly associated with quantitative research, although it is debated in the literature. I find your approach very interesting, particularly in the chapter 'The intervention effect,' which, among other things, addresses the quantitative representation of qualitative results and hypothesis generation. However, please clarify more clearly, what you mean by this in the qualitative context. You often discuss how qualitative approaches enrich quantitative (quasi-experimental) studies, and at times you refer to the 'intervention effect' as something that could potentially replace quantitative approaches. In this context, I would like to refer to the sentence on page 8: 'Note that intervention effects can be evaluated exclusively through qualitative data, or the qualitative data can be an addition to quantitative data within the same study.' There may be a language issue here, but I believe this statement requires further clarification. In my view, qualitative research cannot replace quantitative intervention research, but it can certainly enhance it by providing a clearer understanding of why an intervention works or doesn't work.

• In the ethics statement and data availability section (provided by the submission system), you mention that you obtained permission to describe the data and conduct of the qualitative study by Glebbeek et al. (2024) in this manuscript. You also note an overlap in authorship, which leads me to conclude that some of the authors were involved in both the Glebbeek study and this manuscript. Furthermore, in the manuscript itself, you mention that there was a quantitative study (De Ridder) and a qualitative one (Glebbeek) related to the COVID-19 intervention, with the latter being discussed in this manuscript. That was clear to me. However, the sudden use of phrases such as 'The research goal we defined for the qualitative study…,' as well as 'In our study...' and 'We did...' in the context of the qualitative study was somewhat confusing. I suggest clarifying the distinction for the reader by describing this issue more clearly in the manuscript itself.

• Worked example, page 6: How long, or how many weeks, was an A or B sequence in the A-B design?

• The Table 3 is great because it provides an excellent overview of the topic. One suggestion I have is to introduce an additional column and separate the references into those used for your grounded theory and those that can also be applied (as further reading). Furthermore, I understand that you cannot cover all possible qualitative aspects here, but I would like to point out that qualitative content analysis is also a viable option to be applied in this context.

• Discussion, Limitation, the following sentence: 'Even when invited for the interview near the intervention location, most people preferred the offered option of doing the interview online or at an other time. This changed their status to control group participants.' Please explain why their status was changed to that of the control group.

• Discussion: I would find it interesting if you could revisit and discuss the two research approaches using the studies of Reed 2021 and Geed 2024. You could highlight the benefits that this qualitative study brings to the quantitative results of Reed.

6. PLOS authors have the option to publish the peer review history of their article (what does this mean?). If published, this will include your full peer review and any attached files.

Reviewer #1: No

Reviewer #2: No

---

## [Author Response · Author response to Decision Letter 1]

Please see the uploaded file for a better lay-out.

Reviewer Comment Response

Rev 1 The article submitted for my review is entitled “A qualitative evaluation of an experiment: Studying the effects of empathy-inducing probes on distancing during COVID-19 to derive methodological guidelines”. The article is classically structured and clearly argued. The theme of the article is stimulating: designing a qualitative evaluation method for a quasi-experiment. Nevertheless, I feel that certain elements of the article need to be improved or supplemented. We thank the reviewer for their kind words regarding the theme of the paper, as well as the structure and arguments in the manuscript. Before responding to all of the comments one by one, we have a more general response as well.

From the questions and comments of reviewer 1, it is clear to us that we need to better outline the main goal of this study and paper. By doing that we trust that this also solves many of the more specific comments but of course we will also still address them one by one.

The goal of this paper is to show (quantitative) researchers in the behavioral sciences how to use qualitative methods in the context of experimental research. We use the paper of Glebbeek et al. (2024) as a worked example, which was executed in the larger context of research by De Ridder et al. (2020). In our manuscript, we summarize the steps needed to execute a qualitative analysis in an experimental study. The aim of this paper, therefore, is to serve as a methodological tutorial. It’s developed by describing and discussing the methodological steps and considerations of the empirical example (Glebbeek et al) that is used as illustration throughout the paper.

To bring this structure and the scope of our paper across we made several changes to the manuscript: 1) we changed the title, 2) we have rewritten the abstract, 3) throughout the introduction we rephrased the scope of the paper (mostly on p.2 and p.4-5).

1. In the introduction, despite the authors' didactic efforts, the distinction between experience and quasi-experience is not clear. The authors do not give a clear definition of quasi-experience and what distinguishes it from experience. This is important, however, as the originality of the article seems to rest on the study of a quasi-experience. On p.2, in the second paragraph, we reformulated the definitions of quasi- en true experiments to better clarify the distinction.

As outlined above, we made several changes to better communicate the scope of this article and its originality. Since this is not the presentation of a quasi-experiment we made additional changes in formulations and explanations. In most places, we replaced the term “(quasi-) experiments” (comprising both options) with experimental designs (also comprising both options) and are explicit about when we mean true experiment or quasi-experiment. We added an explanation on p.6, that, while our illustrative example concerns a quasi-experiment, the methodological guidelines we present in this manuscript can also be applied to other experimental designs.

2. Still in the introduction, the authors write “there is little literature on using a qualitative approach for the evaluation of experimental designs in behavioral science.” Firstly, I don't really agree with this peremptory assertion. Numerous books or articles deal to a greater or lesser extent with the qualitative evaluation of experimental design in behavioral science (for example: [1]Kite & Whitley, 2012. Principles of research in behavioral science; [2]Orcher, 2016. Conducting research: Social and behavioral science methods; [3] Maxwell 2008. Designing a qualitative study The SAGE handbook of applied social research methods; etc.). Secondly, there are a multitude of methods for evaluating qualitative experiments ([4]Rossman & Wilson, 1985, Numbers and words: Combining quantitative and qualitative methods in a single large-scale evaluation study. Evaluation review, 9(5), 627-643; [5] Horsburgh, 2003, Evaluation of qualitative research. Journal of Clinical Nursing (Wiley-Blackwell), 12(2); [6] Hennink, et al (2020). Qualitative research methods. Sage; [7] Yadav, 2022. Criteria for good qualitative research: A comprehensive review. The Asia-Pacific Education Researcher, 31(6), 679-689. etc.).

2B. I fail to see where the contribution of the new method proposed by this article lies. It should be clarified in a very precise and documented way. In particular, it should be specified whether the proposed method concerns quasi-experiments or experiments, whether it concerns mixed or totally qualitative methods, or even experiments that are mainly quantitative but incorporate qualitative data; otherwise, it's hard to see where the article's contribution lies. At the end of the introduction, the authors state “we provide a step-by-step approach step-by-step approach for future researchers wishing to carry out a qualitative evaluation of a"quasi-)experimental design”, but it is not clear whether this is really a research gap. In this respect, in the “Methodological approach” section, the authors state “Our own study is an example of adding a qualitative element to a quantitative study”, so we'd be tempted to believe that the gap lies here, but we'd have to justify it much earlier in the article. Thank you for providing suggestions for relevant literature. We were familiar with almost all, and checked the other references. While this work is indeed about qualitative research, it lacks the specific methodological focus on the qualitative evaluation of experimental designs in behavioral sciences.

We added several of the suggested references for further reading on qualitative methods in general (p.4). Also the references about quantifying qualitative results that were suggested by the reviewer, are now included (p.4).

2B. We trust that with a better presentation of the scope of this paper (see first response), and a better positioning with regards to type of designs (see response to remark 1), we have now clarified what the contribution of this paper is.

3. Still in this “Methodological approach” section, the authors mention a large number of bibliographical references (references 20 to 61). However, in a research article, bibliographical references usually appear in the introduction or discussion, but not in the methodological or results section. This gives the impression that your work is more like a literature review than a concrete experiment. The authors should clearly state their position on this point. Is the article a methodological review or the testing of a qualitative experimental method? This is not very clear. Thanks again for pointing out that the scope of the paper was really not clearly presented. We prefer the term methodological tutorial over methodological review but, indeed, the main message is not about the results of the illustrative example we used; and thus the manuscript does not follow the typical structure of a paper presenting empirical research. See also our first general response.

With regards to questions 3, 4 and 5: We do realize that we were also using confusing language to refer to the different studies that play different roles in this work. This manuscript presents a methodological tutorial. The worked example, is the qualitative study that evaluates a quasi-experiment presented in Glebbeek et al. The study by Glebbeek et al. is part of a larger study presented by De Ridder. The different studies are now explicitly and consistently referred to as, respectively, “our study” (the tutorial), the worked example/illustration/Glebbeek et al. (the empirical qualitative study) and the bigger context study/De Ridder et al. (providing the study design and experimental manipulation).

4. In the “data collection” section, you state that you have carried out interviews and observations, but it's hard to understand why you are doing so (for what purpose) and with whom (list of interviewees). What's more, if I understand correctly, the purpose of these observations and interviews is to justify the results of the method you describe, but your results contain very little verbatim information about these observations and interviews. So I don't see how these observations and interviews validate your method. We feel that this remark follows from the same misconception about the general scope and the role of the empirical example in this paper. In the paper by Glebbeek et all., the empirical study and its results are presented in detail. Here, we just use it as an illustration to present a (relatively unknown and not often used) method. See also response 3.

As research methodologists with expertise and experience in qualitative methods, we feel we are in the position to write this tutorial, outlining how experimental designs can be evaluated using qualitative data and analysis. The aim is not to validate a method through an example study.

5. Still in the presentation of the method, you insert a few verbatims after figure 2 and then announce that “Observations and interviews clearly showed that social interactions are context-dependent, i.e. that behavior depends not only on intentions (based on reasoning about actions and motivations for complying with them) but also, for example, on how friends or other contacts behave, their intentions and the perceived risk of not behaving as intended (see figure 2).”. It's possible to agree with this statement, but I don't see how it validates your method of evaluating a qualitative experiment? In my opinion, to validate a method, it is necessary to provide very factual evidence, for example by comparing the responses of the experimental group and the control group, and by showing that each stage of the method improves the results. In my opinion, your article fails to do this. Table 2 compares the results between the two groups, but I fail to see how this validates your method. To validate your method, it would probably have been more appropriate to set up two groups, an experimental group studied using the new method, and a control group not studied using this method. We assume this comment is resolved by the changes as discussed in our first general response, combined with the responses for 3 and 4.

6. To conclude the presentation and justification of your method, you write that “Our intention was to demonstrate the iterative process of hypothesis generation and validation through repeated re-evaluation of the data.

of data.” Firstly, I don't think this intention corresponds to the initial objective of your study set out in the introduction. Secondly, I discover that your method ultimately consists in determining hypotheses by re-evaluating data. Figure 1, which presents your method, and Table 3 do not mention the construction of these hypotheses, which, in my opinion, is part of a qualitative abductive research methodology not presented in this work. We thank the reviewer for pointing out the confusing use of the word hypothesis in this part of the manuscript. To better convey our message we replaced the term ‘hypothesis’ with ‘preliminary explanation’, as we agree that ‘hypothesis’ can be misleading in this context. The iterative process of formulating ideas on potential themes and relations between themes, and checking/testing those ideas with the other available data is inherent to the processes of axial and selective coding, both included in Table 3.

By replacing the term, and defining the replacing term on p.26, we trust this point gets across more clearly now.

7. I'm not very convinced by your discussion, as you fail to demonstrate the validity of your method. You write “Table 3 summarizes the essential elements for conducting a rigorous qualitative investigation of potential research projects.” Table 3 presents a literature review of the various stages (Research objective; Data collection

Personally, I regularly conduct qualitative studies and apply these steps from the literature, but I've never tested the validity of the overall method, and I don't know if the method is valid. After reading your work, I still don't know. You have carried out a literature review of the design stages of a qualitative study, which is interesting, but it is not possible to say, as you do, that this method is rigorous and reproducible. To write that, it would have been necessary, at the very least, to apply this method to several studies, or to analyze the results between an experimental group and a control group. I'm not saying that your method isn't correct, but you don't provide proof of it. See our earlier responses, especially for comment 4. Testing the validity of a method is not the scope of this paper. We aim to provide a methodological tutorial on applying qualitative methods in the context of experimental designs.

Rev 2 1. Abstract: Please include the qualitative research design, specifying that the qualitative evaluation was conducted using Grounded Theory. I would also suggest clarifying this earlier in the manuscript to help guide the reader (e.g., a guide for a qualitative evaluation based on Grounded Theory). Thank you for pointing this out. We agree and added this to the abstract (p.1) and in the introduction at an earlier stage (p.4).

2. A. In your manuscript, you frequently refer to the 'effect of the intervention' or 'intervention effect' in the context of the qualitative approach. The term 'effect' is more commonly associated with quantitative research, although it is debated in the literature. I find your approach very interesting, particularly in the chapter 'The intervention effect,' which, among other things, addresses the quantitative representation of qualitative results and hypothesis generation. However, please clarify more clearly, what you mean by this in the qualitative context.

2B. You often discuss how qualitative approaches enrich quantitative (quasi-experimental) studies, and at times you refer to the 'intervention effect' as something that could potentially replace quantitative approaches. In this context, I would like to refer to the sentence on page 8: 'Note that intervention effects can be evaluated exclusively through qualitative data, or the qualitative data can be an addition to quantitative data within the same study.' There may be a language issue here, but I believe this statement requires further clarification. In my view, qualitative research cannot replace quantitative intervention research, but it can certainly enhance it by providing a clearer understanding of why an intervention works or doesn't work. 2A: We used the term intervention effect to describe any change that is assumed to be the consequence of being exposed to an intervention/manipulation, irrespective of how this change is defined or measured (i.e., with quantitative outcomes/variables or in a qualitative way). However, we agree that readers may interpret the word ‘effect’ only as a quantitative measure. We therefore added such an explanation on p.3.

2B: First of all, we see how our phrasing could be confusing, and adjusted it (p. 8). Obviously, we agree with the reviewer that qualitative experimental research cannot in general replace quantitative experimental research. However, we do believe and argue that one could, on occasion, decide to focus solely on qualitatively defined and measured effects.

3. In the ethics statement and data availability section (provided by the submission system), you mention that you obtained permission to describe the data and conduct of the qualitative study by Glebbeek et al. (2024) in this manuscript. You also note an overlap in authorship, which leads me to conclude that some of the authors were involved in both the Glebbeek study and this manuscript. Furthermore, in the manuscript itself, you mention that there was a quantitative study (De Ridder) and a qualitative one (Glebbeek) related to the COVID-19 intervention, with the latter being discussed in this manuscript. That was clear to me. However, the sudden use of phrases such as 'The research goal we defined for the qualitative study…,' as well as 'In our study...' and 'We did...' in the context of the qualitative study was somewhat confusing. I suggest clarifying th

---

## [Decision Letter · Decision Letter 1]

3 Apr 2025

PONE-D-24-29638R1Applying qualitative methods to experimental designs: A tutorial for the behavioral sciencesPLOS ONE

Dear Dr. Leplaa,

Thank you for submitting your manuscript to PLOS ONE. After careful consideration, we feel that it has merit but does not fully meet PLOS ONE’s publication criteria as it currently stands. Therefore, we invite you to submit a revised version of the manuscript that addresses the points raised during the review process. I, as well as the reviewer, appreciate all the effort that went into addressing the comments in the first round. It is clear that much of this was taken into account and integrated into the manuscript. Still, there is a major concern the reviewer and I both share in terms of what is not included. Specifically, it is impossible to assess the quality, reliability, and validity of the qualitative method proposed by authors without presenting the data used to develop and test this method. Much more is needed in this regard, as the reviewer lays out. As a note, unfortunately, if it is not addressed in this round, the paper will most likely be rejected.

We look forward to receiving your revised manuscript.

Kind regards,

Gabriel Velez, Ph.D.

Academic Editor

PLOS ONE

Reviewers' comments:

Reviewer's Responses to Questions

**Comments to the Author**

1. If the authors have adequately addressed your comments raised in a previous round of review and you feel that this manuscript is now acceptable for publication, you may indicate that here to bypass the “Comments to the Author” section, enter your conflict of interest statement in the “Confidential to Editor” section, and submit your "Accept" recommendation.

Reviewer #1: (No Response)

2. Is the manuscript technically sound, and do the data support the conclusions?

Reviewer #1: No

3. Has the statistical analysis been performed appropriately and rigorously? 

Reviewer #1: No

4. Have the authors made all data underlying the findings in their manuscript fully available?

Reviewer #1: No

5. Is the manuscript presented in an intelligible fashion and written in standard English?

Reviewer #1: Yes

6. Review Comments to the Author

Reviewer #1: Dear authors,

I would like to thank you very much for the efforts you have made to rework and revise your article. The objective of the article is much clearer and the concepts are much better defined. To further improve this work, I recommend that the authors take the following remarks into consideration:

1) The objective of the article is now clearly defined: “to propose a step-by-step approach for future researchers who wish to conduct a qualitative evaluation of a (quasi-)experimental design”. However, in the title of the article, but also in the text, it is difficult to understand whether the qualitative evaluation proposed relates to an experiment or a quasi-experiment. The authors would do well to be consistent throughout the article and to refer to either qualitative evaluation of experiment or qualitative evaluation of quasi-experiment.

2) Also in the introduction, I understand that the research gap is as follows: “there is little literature on the use of a qualitative approach for the evaluation of experimental models in the field of behavioral sciences”. Therefore, is it not necessary to specify in the title of the article and in the research objective that it is a qualitative evaluation in the field of behavioral sciences? Moreover, on pages 6 and 7, the authors again indicate their research objective and mention that the objective is “the study of how to conduct a qualitative study of a (quasi-)experiment in the context of behavioral research.” In my opinion, it is necessary to be consistent and to follow the same research objective throughout the article.

3) In the methodological presentation, the authors first present the previous work of De Ridder et al. and Glebbeek et al., which seem to be the work on which the methodological study of this article is based.

Therefore, it would be appropriate to present the samples of each of these two studies in detail. How many people did these studies cover? Who were the respondents and interviewees? How were the control and intervention groups, if any, constituted? Throughout the methodological presentation that follows, the characteristics of the samples and data that were studied are not known. You present the different stages of the qualitative evaluation but we never know which people were questioned or interviewed, how these people were chosen, or how the control and intervention groups were designed. This is very problematic because we want to believe you, but it is difficult to evaluate the relevance of a qualitative method if you do not present the data used.

4) In your proposed evaluation method (figure 1), you emphasize data collection and analysis as essential steps, but in your own study you do not present this data and you do not indicate how it was collected. Once again, I understand that you have used secondary data from the work of De Ridder et al and Glebbeek et al. but you must present the characteristics of this data otherwise it is very difficult to judge the quality of your study and the qualitative method you propose. It is not a question of giving us access to all the data but of presenting it. When were they collected? From whom? How did the interviews go? What were the socio-demographic characteristics of the interviewees? You state on page 7 that “The opening up of data is also an essential aspect of scientific research, as it reinforces transparency, responsibility and reproducibility”. However, you do not present the data from your study. This is rather problematic.

5) Tables 1 and 2 are interesting because they present the data coding elements, but we still do not know the data used (table 1) and the characteristics of the people interviewed; nor do we know the make-up of the control and intervention groups (table 2). Without a presentation of the socio-demographic characteristics of these data, for my part, it is impossible for me to say whether your evaluative method is reliable and valid. In my opinion, no one can comment on the validity and reliability of a method without knowledge of the data used to test that method. Despite all the kindness that one can have for your study, it is impossible.

6) The discussion and contributions are interesting but once again, without a detailed presentation of your data, it is not possible to tell you whether or not your contributions and your 4-step methodology are valid. Furthermore, you state that the quantitative method reinforces the quantitative methods but it is not very clear, if this is the case, how the qualitative data of Glebbeek et al. reinforce the quantitative data of De Ridder et al.

I therefore recommend that you present your secondary data from the two previous studies in detail (in one or two additional tables, for example). Without the presentation of this data, I cannot comment on the results of your study and the reliability of your method, even if I can assume that it is sound. I am not asking for the identity of the people interviewed or questioned, but simply for their number, their socio-demographic characteristics, the way in which they were divided into the control and test groups, the duration of the interviews, the interview guide and any other relevant information that would enable the reliability and validity of your qualitative method to be evaluated. Thank you for your understanding.

Thank you very much for giving me the opportunity to read your work, which has a potentially very interesting objective.

Best regards

7. PLOS authors have the option to publish the peer review history of their article (what does this mean?). If published, this will include your full peer review and any attached files.

Reviewer #1: No

---

## [Author Response · Author response to Decision Letter 2]

See the cover letter for a better edited version of this response:

We understand the confusion created by using a quasi-experiment as the worked example, while the methodological tutorial addresses experimental designs in general (including true experiments and quasi-experiments, as specified on lines 10-11). However, the methodological steps presented in this tutorial are equivalent for the two types, so it does not make sense to restrict the general presentation to quasi-experiments only, just because our worked example is of that type. We understand that this needs explicit communication and consistency in language throughout the paper and we thank the reviewer for pointing out that this needs improvement.

On lines 20-22, we added an explicit statement: “Throughout the paper, when we use the term ‘experiment’ or ‘experimental design’ this includes both true and quasi- experiments.”

We trust that by adding this explanation, this avoids the potential ambiguity at several positions in the paper, e.g., lines 84-86 state: “We provide a step-by-step approach for future researchers aiming for a qualitative evaluation of an experimental design, supported with a worked example.” Also, in the title we use the term “experimental designs” and not experiment or quasi-experiment.

In the presentation of the worked example, on lines 136-138 we state explicitly: “While the GLS study is an example of a quasi-experiment, the lessons and guidelines can be applied directly to true experiments as well.”

To further avoid confusion, throughout the paper we have removed every use of the term “(quasi-)experimental” and changed this into “experimental design” (lines 60, 135, 169, 451).

We agree with the reviewer that the title should capture exactly that. However, in our opinion, it does, because the title states: “Applying qualitative methods to experimental designs: A tutorial for the behavioral sciences.”

We are therefore not sure what the reviewer is missing (or perhaps the change in title between version 1 and 2 of the manuscript was not noticed?). If we are misunderstanding this remark, and it is not the term behavioral that is considered problematic but instead it is another reference to the confusion we created with the term (quasi-) experiments, then we refer to our response 1.

Our methodological tutorial is indeed based on the empirical study of Glebbeek et al., but not directly on the study by De Ridder et al. What we aimed to communicate is that we made use of the intervention that De Ridder et al. set-up for their own study, but other than that the Glebbeek study was executed independently from the De Ridder study. We made the following changes to better explain this in our manuscript:

On lines 86-100 we rephrased the introduction and description of both studies and their relation to the methodological tutorial presented in this manuscript.

We also changed the section title “Worked example” into ‘Context of the worked example’ in order to better represent the goal and content of this section. The section describes the study design and interventions of the De Ridder et al. study, because Glebbeek made use of this set-up, but the De Ridder study itself is not our worked example (Glebbeek et al. is).

We seriously considered if presenting information about the De Ridder study (e.g. sample characteristics) would be a useful addition to this paper and decided against it. The data collected by De Ridder et al. is not used in this paper; the analyses we use as illustrations in this paper are only based on the Glebbeek study.

We do understand the request for additional information about the Glebbeek study and have provided much of the requested information.

We have added sample information in Section 2.2: Data collection. On lines 294-296 we give more information on who were observed, and on lines 331-334 we give more information on who were interviewed.

We also added appendices A through D, containing the observation schemes and topic lists used for data collection by Glebbeek et al. We introduce these appendices on lines 292, 302, and 324.

In Section 2.3: Reflecting and Adjusting, we shortly discuss the theoretical sampling plan (lines 404-413) and explain that we limited the amount of information collected on participants, in order to prevent undue intrusion in their privacy. Collecting qualitative data always means that you collect quite some personal data, and limiting the amount of data to only the relevant information protects your participants from possible harm. With that in mind, Glebbeek et al. stopped registering, for example, gender and age and other demographic information that deemed not relevant in this study.

See our responses to 3/4 and corresponding additions to the paper.

In addition, we also like to stress again that the worked example is not used to validate the method but to illustrate the methodological steps presented in the tutorial. As such, the tutorial is based on the combination of a review of literature on qualitative research in the context of behavioral sciences, and our extensive personal experiences with qualitative research in general, and in this illustrative example evaluating results from an experimental design in particular. We very carefully checked that we do not claim, anywhere in the paper, that illustrations from the worked example are presented as evidence of validity.

Also, we like to refer to lines 841-848 (last paragraph of the discussion) for a modest formulation of what we offer: ‘a useful starting point, without claiming

that the presented guidelines are all-encompassing.’

---

## [Editor Report · Decision Letter 2]

4 May 2025

Applying qualitative methods to experimental designs: A tutorial for the behavioral sciences

PONE-D-24-29638R2

Dear Dr. Leplaa,

We’re pleased to inform you that your manuscript has been judged scientifically suitable for publication and will be formally accepted for publication once it meets all outstanding technical requirements.

Kind regards,

Gabriel Velez, Ph.D.

Academic Editor

PLOS ONE
---

## [Editor Report · Acceptance letter]

PONE-D-24-29638R2

PLOS ONE

Dear Dr. Leplaa,

I'm pleased to inform you that your manuscript has been deemed suitable for publication in PLOS ONE. Congratulations! Your manuscript is now being handed over to our production team.

Kind regards,

on behalf of

Dr. Gabriel Velez

Academic Editor

PLOS ONE